# Religious Coping Styles and Depressive Symptoms in Geriatric Patients: Understanding the Relationship through Experiences of Integrity and Despair

**DOI:** 10.3390/ijerph19073835

**Published:** 2022-03-23

**Authors:** Lindsy Desmet, Jessie Dezutter, Anne Vandenhoeck, Annemie Dillen

**Affiliations:** 1Faculty of Theology and Religious Studies, KU Leuven, 3000 Leuven, Belgium; anna.vandenhoeck@kuleuven.be (A.V.); annemie.dillen@kuleuven.be (A.D.); 2Faculty of Psychology and Educational Sciences, KU Leuven, 3000 Leuven, Belgium; jessie.dezutter@kuleuven.be; 3Faculty of Theology, North-West University, Potchefstroom 2520, South Africa

**Keywords:** religious coping, depressive symptoms, geriatric patients, integrity and despair, old age

## Abstract

Older persons are often confronted with challenging events in their lives. Religion can offer them a way to deal with these challenges. The study of religious coping styles helps us to understand how people find support in their religion or wrestle with aspects of their religion when they are confronted with difficulties. Especially when older adults face illness and hospitalization, religious coping styles might be triggered. Despite the fact that the public role of religion, especially Christianity, is diminishing in West European societies, a large group of Belgian geriatric patients call themselves religious. Previous studies have shown that there is a link between positive/negative religious coping styles and the depressive symptoms that often occur in older adults. More recently, some scholars have emphasized that this relationship is more complex. Therefore, this paper investigates the role of one possible underlying mechanism between positive/negative religious coping styles and depressive symptoms in geriatric patients, namely the developmental process of integrity and despair as two factors within this mechanism. One hundred thirty-nine geriatric inpatients from three hospitals in Belgium who reported to feel religiously affiliated were involved in this study. Our results indicate that experiences of integrity and despair function as an explanatory pathway in the relationship between negative religious coping styles and depressive symptoms. Further, a direct link was found between both when accounting for experiences of integrity and despair. For positive religious coping styles, no direct or indirect relationship with depressive symptoms was found. In healthcare, geriatric caregivers need to be aware of the interaction between positive and negative religious coping styles, the developmental process of integrity and despair, and depressive symptoms.

## 1. Introduction

### 1.1. Positive and Negative Religious Coping Styles

When people face life situations that they appraise as stressful events, emotions of distress can emerge. To deal with these situations and emotions, people make use of coping mechanisms [1]. Richard Lazarus and Susan Folkman [2] (p. 141), who developed the stress and coping theory, define coping as “constantly changing cognitive and behavioral efforts to manage specific external and/or internal demands that are appraised as taxing or exceeding the person’s resources”. According to Lazarus and Folkman, people can use various kinds of coping styles, which reflect different ways of dealing with the stressful situation, such as seeking social support, emotion-focused coping styles such as emphasizing the positive and making use of self-blame, and problem-focused coping styles [3].

Further, religion can play an important role in coping [4]. According to Crystal Park [5], religion can be understood as a possible meaning system through which people experience their environment. Consistently, religious coping functions as a religious meaning system that can help people cope with stressful events by providing a meaning-making framework [5,6]. Religious coping can function as an established coping style in religious people and can play a role in the way people reappraise illness, loss, suffering, and adverse moments in life [7,8]. Mostly, religious coping is found to be a helpful strategy, providing support, strength, and comfort during challenging moments in life [9]. Harold G. Koenig [7] (p. 285) states that religious beliefs can be a helpful coping mechanism because they “provide a sense of meaning and purpose during difficult life circumstances”. Nevertheless, religious coping is not always helpful [10,11]. For example, serious illness may be accompanied by feelings of disappointment in religious belief systems, by religious distress, and by the use of punitive religious reappraisals [12].

To meet the various forms of helpful and harmful religious coping styles, a distinction has been made in research between positive and negative religious coping styles [13,14]. Both styles differ in the way people perceive religion and religious communities, and especially in their interpretation of their personal relationship with God [15]. Positive religious coping yields a trusting and secure relationship with a loving and supporting God. People maintaining positive religious coping feel harmoniously connected with God and sustain a collaborative relationship with God [16,17,18]. In most studies, people report higher levels of positive religious coping styles compared to negative religious coping styles [12,19,20]. Negative religious coping reflects a distant and negative appraisal of God based on a conflicting relationship with God, including feelings of being punished by God and desolation [16,20]. Sometimes, negative religious coping styles manifest as religious struggles or doubt, which encompasses a possibility for religious growth, transformation, and healing towards a more positive incorporation of religious coping in the long term [16,21,22,23]. An example of a religious struggle is questioning if God is still a supporting and reliable God. Although the distinction between positive and negative religious coping styles suggests that they are opposites, and thus related, they need to be considered as separate constructs. This idea has been confirmed by previous studies reporting no significant correlation between positive and negative religious coping [20,24,25,26,27]. More concretely, the use of positive religious coping does not simply result in a lack of negative religious coping, and vice versa [28,29]. For example, it is possible that a patient mainly uses positive religious coping styles and simultaneously struggles with his or her belief in God.

Two factors influence the role of religion and religious coping styles in late life in West European Societies. On the one hand, religion and religious coping are more prevalent among older people compared to younger people [30,31,32]. In 2018, The Pew Research Center reported that around the world, people in their forties and older are more religious than younger adults. Especially in societies that are predominantly Christian, such as West European countries, a so-called age gap arises [33]. In addition, old age involves various kinds of loss, suffering, and illness, triggering the use of coping systems, which also can imply an increase of religious coping strategies in religious older adults [20,34]. On the other hand, the public role of religion is changing in secularized countries, and this might also affect the role of religious coping in aging. Today, in West European societies, most older adults were raised in predominantly Christian societies and have experienced the shift from a society in which Christianity was common and public to one in which the public role of Christianity has declined, and religious practices have become more private. In Belgium, more than half of the country (56%) call themselves Christian (most Belgian Christians are Roman Catholics), but the majority of this group (46%) are non-practicing Christians [35]. Only a minority (10%) is involved in Church activities. Further, 38% feel religiously unaffiliated, and 7% indicate having another religion (mostly Muslim) or did not know what to answer or refused to answer.

It is important to take this context into account when investigating people’s religious coping styles. Based on this information, it is of interest to what extent religious coping is incorporated in geriatric patients’ life in Belgium. In this study with Belgian religious geriatric patients, religious coping is perceived from the Roman Catholic tradition.

### 1.2. Positive/Negative Religious Coping Styles and Depressive Symptoms

The relationship between positive and negative religious coping styles and depressive symptoms is particularly of interest because of the increasing risk for depressive symptoms in late life [20,36]. Depression is considered one of the most common mental illnesses in late life [37,38]. Among others, feelings of loneliness, lack of social support, negative life events, loss of loved ones, living alone, frailty, and illness or disability are aspects inherent to late life-triggering depressive symptoms [39,40,41,42,43]. In geriatric patients living in Europe or in the United States, prevalent rates of depressive symptoms vary widely from 10% to 65% [13,44,45,46,47]. For example, a recent study in the Netherlands with acutely hospitalized older adults reported that 36.43% had mild or severe depressive symptoms [48].

In most studies conducted with different age groups, negative religious coping styles are positively related to depressive symptoms [20,49,50,51,52,53,54]. However, that does not imply that negative religious coping styles are always related to depressive symptoms [55]. As revealed in Pargament’s study with medically ill elderly patients [50], only people who incorporated negative religious coping styles in a chronic way were at risk of depression. In line with this, a study involving adults with cystic fibrosis showed that divine spiritual struggle was a risk factor for depressive symptoms, but divine punishment was not linked to depressive symptoms [10]. According to positive religious coping styles, these are only occasionally negatively related to depressive symptoms [10,24,55,56,57,58]. For example, in Terreri and Glenwick’s study [26], positive religious coping was not associated with depressive symptoms when controlling for general coping. Further, in a study on religious coping styles among caregivers of terminally ill cancer patients, negative religious coping was related to major depressive disorder, unlike positive religious coping [54]. These findings were also confirmed in studies with other religious groups such as Muslims [59,60] and Jews [61].

The limited number of studies available on older adults’ religious coping strategies confirms the trend that positive religious coping is negatively or not related to depressive symptoms, whereas negative religious coping is, in general, positively related to depressive symptoms [13,24,50,58,62]. In line with what was stated earlier, the common findings in religious coping literature in late life need to be nuanced. For example, in Bosworth and colleagues’ cohort study with depressed older adults in the United States [55], negative religious coping was associated with increased depressive symptoms at the baseline, but not at a follow-up at six months. Surprisingly, positive religious coping did have the capacity to predict depressive symptoms both at the baseline and the follow-up level [55]. This is contrary to the premise that negative religious coping styles, but not positive religious coping styles, are related to depressive symptoms. Despite these worthwhile insights, results on the relationship between positive/negative religious coping styles and depressive symptoms in geriatric patients remains scarce and slightly unclear. This study therefore investigates the relationship between positive and negative religious coping styles and depressive symptoms in geriatric inpatients in more depth and incorporates potential explaining pathways.

### 1.3. The Role of Integrity and Despair

Based on the mixed findings regarding the relationship between positive/negative religious coping styles and depressive symptoms, it is likely that this relationship is more complex. Other factors might mediate the relationship between religious coping styles and depressive symptoms [4]. Ai and colleagues [27] showed that there are psychosocial factors underlying this relationship, such as hope and social support. Similarly, a study by Pearce et al. [54] revealed that social support, optimism, and self-efficacy mediated the relationship between negative religious coping and satisfaction and quality of life. More generally, Koenig argues that, among other factors, social support and cognitive appraisal of stressors mediate the relationship between the use of religion and mental health outcomes such as depression [4]. In other words, there are some indications that the relationship between positive/negative religious coping styles and depressive symptoms is statistically mediated by other factors. In the process of aging, it is of benefit to investigate the role of one of the core processes in late life, namely the way people reflect back on their lives and how they perceive the value of their past life. This process in late life is described by Erik Erikson [63].

According to Erikson, the psychosocial development of people’s identity consists of eight stages, from birth to old age [63,64]. Each stage consists of a specific developmental task related to a psychosocial conflict. Successfully dealing with this psychosocial conflict means finding a balance between two opposing components that are specific to the developmental stage of life [63]. The eighth stage of Erikson’s psychosocial development theory, which is the stage of late life, is described as an introspection and reflection on past life, and it goes along with the psychosocial crisis of experiences of integrity and despair [63,64]. When people have a positive reflective life process, experiences of integrity are present. When individuals are able to accept failures, come to terms with disappointments, forgive others and oneself, reflect on the own life in a kind and mild manner, the psychosocial crisis will result in feelings of a meaningful and satisfying life, despite the potential hurt and sorrow [64,65]. When people are, however, dissatisfied with their past life, struggling with regrets, failures, and disappointments in life, and are not able to integrate the negative life aspects into the whole life experience, feelings of despair arise. In this case, individuals realize that they are not able to restart their lives and solve all the missed chances or restore broken relationships. They feel unable to accept and forgive, which results in feelings of sorrow, pain, and despair [64,65]. According to Erikson, the key task in late life is to strive for a deliberate and balanced reflection between experiences of integrity and despair [64]. A disbalance between experiences of integrity and despair occurs when someone experiences difficulties with accepting the past, does not perceive their life as meaningful, and has a predominantly negative view on life [66]. In this case, the psychosocial crisis remains unresolved.

Previous studies suggested a significant relationship between experiences of integrity and despair and depressive symptoms. For example, Dezutter and colleagues [67,68,69] indicated, in a sample of nursing home residents and community-dwelling older adults, that high experiences of despair were related to high levels of depressive symptoms, and high levels of integrity were related to low levels of depressive symptoms. In another study with a Dutch sample of 218 older adults, depressive symptoms were positively related to despair [70]. However, the relationship turned out to be non-significant when controlling for personality traits. In Rylands and Rickwood’s study with older women [71], it was revealed that not being able to accept the past was significantly related to depressive symptoms. Lastly, van der Kaap-Deeder et al. [72] recently found that integrity and despair were, respectively, negatively and positively related to depressive symptoms in older adults. In our study, we assume that the way geriatric patients reflect on their life in terms of experiences of integrity and despair is linked with current depressive symptoms during hospitalization.

In addition, as Park [6] describes religious coping as a meaning-making process, it might be that the helpful and harmful aspects of, respectively, positive and negative religious coping styles are related to the way people value their past life. For example, as stated by Koenig [73] (p. 76), positive religious coping styles can bring in the perspective of a “next life” with new opportunities and provide a framework for the forgiveness of failures in one’s current or past life. However, negative religious coping styles may also be harmful in relation to the psychosocial crisis in late life. For example, the principle of sinfulness in Christianity, as well as God’s judgment, can make it difficult to come to terms with failures or disappointments in one’s past life. We assume that the specific coping style that one uses might be related to how this psychosocial crisis is handled. This might especially be the case for geriatric patients who are confronted with this psychosocial crisis in a hospital situation that often triggers uncertainty, pain, and sorrow. Since we know that older adults still regularly use religious coping styles, we wonder how these religious coping styles are related to the psychosocial crises they are facing in this life stage.

In sum, as experiences of integrity and despair are dynamic core processes specific to late life, it is important to investigate whether this developmental process of integrity and despair plays a role in the relation between established positive/negative religious coping styles and depressive symptoms.

### 1.4. Aim of the Study

In this study, we investigate three hypotheses. The first two seek to replicate previous findings from international studies specifically in Belgian geriatric patients, and the third wants to extend these findings.

Firstly, based on previous findings that styles of religious coping are related to the absence or presence of depressive symptoms, it is our aim to replicate these findings in geriatric patients. We hypothesize that positive religious coping is negatively related to depressive symptoms, and negative religious coping is positively related to depressive symptoms.

Secondly, we want to investigate whether experiences of integrity and despair are related to depressive symptoms in Belgian geriatric patients during a hospital stay, in line with earlier findings in late-life literature. We suppose that the experiences of integrity and despair will both be related to depressive symptoms in a negative and positive way, respectively.

Thirdly, and more exploratory, we want to extend current research insights by clarifying whether the psychosocial crisis of integrity and despair plays a role in the relationship between positive/negative religious coping styles and depressive symptoms. We hypothesize that positive/negative religious coping is related to depressive symptoms through its relationship with experiences of integrity and despair. Since this mediation hypothesis is rather an exploratory investigation, we do not know the direction in which the relationship will manifest.

## 2. Materials and Methods

### 2.1. Procedures and Data Collection

This cross-sectional study is part of a larger study (preregistration of the study: https://osf.io/wdnq6/) on the relationship between geriatric patients’ spiritual needs and ill-being [74]. This study involves three hospitals in Flanders, Belgium, with a total sample of 201 geriatric patients. For this study, we analyzed the data collected from June to August 2020 from 139 geriatric patients. The patients included in this study were selected if they stated that they were believers and felt affiliated to a religious tradition. As the questionnaire on religious coping includes specific items on religion, people who reported being non-believers, or believers without religious affiliation, were excluded from this part of the study. In total, 145 out of 201 participants (72.2%) reported being believers and feeling religiously affiliated. However, as the questions on religious coping were at the end of the questionnaire, six participants were too exhausted to answer the questions on religious coping and dropped out.

Patients were included in this part of the study if they were over 65 years old, stayed in the hospital, were capable of understanding and answering the Dutch questionnaire and giving informed consent, and were religiously affiliated believers. If patients experienced severe mental health problems, severe forms of memory loss, or were mentally disabled or terminally ill, they were excluded from the study. Both the Ethics Committee Research UZ/KU Leuven (ID S63617) and the local Ethics Review Board of each hospital approved the study. Patients did not receive compensation for their participation in the study.

All geriatric patients were interviewed in their hospital rooms. Based on the inclusion and exclusion criteria, the head nurse informed the researcher about eligible patients. After verbally informed consent was obtained, a standardized interview with closed-ended questions was carried out by the first author of this study. The researcher registered the answers in the online Qualtrics software tool.

### 2.2. Participants

Participants (*n* = 139) were aged between 70 and 100 years with an average age of 85.13 (*SD* = 5.49), of which 69.8% were female. Most participants were widowed (57.6%), 33.1% were married, 7.2% were single, 1.4% were divorced, and 0.7% were in a relationship. About half of the patients (51.1%) received secondary education up to fourteen/sixteen years old. Of the others, 2.9% obtained primary school degree, 29.5% received secondary education until eighteen years old, and 16.5% achieved a university or college degree. All participants had Belgian nationality. Most patients stayed for a couple of days or one week in the hospital (74.8%), some for two or three weeks (14.4%), and a minority for more than three weeks (7.2%). Additionally, 3.6% did not report their length of stay in the hospital. Only seven patients were hospitalized in a rehabilitation ward, and the remaining majority were in geriatric wards.

Most participants called themselves Catholic (96.4%), 2.9% indicated being Christian, and 0.7% were Muslim. Further, 41.8% reported that religion was very important in their lives, whereas 33.8% experienced religion as important, 16.5% described religion as somewhat important, and 7.9% said it was not important in life. More than half of the participants prayed on a daily basis (56.1%), 2.9% prayed on a weekly basis, 23.0% prayed occasionally, 5.8% prayed seldom, and 12.2% never prayed. Finally, 47.5% of the sample were involved in religious activities on a weekly basis (one person in this group reported being involved in religious activities every day). The others were less frequently involved, namely 25.2% occasionally, 14.4% seldom, and 12.9% never (“A lot of participants that were never, rarely or occasionally involved in religious activities, noticed that they were not able to attend religious activities, because of their physical disabilities” [74] (p. 7)).

### 2.3. Measurements

#### 2.3.1. Positive and Negative Religious Coping

Positive and negative religious coping strategies were assessed with the 14-item Brief Religious Coping Scale (Brief RCOPE) developed by Pargament [16,18], consisting of the positive and negative religious coping scale. Both scales are measured separately and include seven statements that can be answered with totally not (1), rather not (2), rather yes (3), and totally yes (4). The positive religious coping scale consists of items such as “trying to see how God might be trying to strengthen me in this situation” and “asking for forgiveness for my sins”. The negative religious coping scale is comprised of questions such as “wondering whether God had abandoned me” and “questioning the power of God”. The questionnaire has been used in previous studies with older adults showing good psychometric properties [16,20,75]. The positive and negative religious coping scales have good internal consistency in the current sample with, respectively, Cronbach’s α = 0.86 and Cronbach’s α = 0.71.

#### 2.3.2. Integrity and Despair

To assess experiences of integrity and despair, the shortened version of Van Hiel and Vansteenkiste’s validated Dutch questionnaire [76] was used with four (integrity) and five (despair) items, ranged on a 4-point Likert scale from 1 (totally disagree) to 4 (totally agree). The subscale of integrity contains four items, such as “I can accept the bad moments of my past life” and “I can give negative past experiences a place in my life”. The subscale of despair includes five items, such as “I look back upon my life with a feeling of discontent and regret” and “I wish I had lived my life differently”.

One item of the despair scale (“I have trouble accepting the idea that what has happened is largely the result of what I have done myself”) was experienced by the participants as too difficult to understand. Even if participants tried to answer the question, they mentioned that they were not sure if they understood the question correctly. As participants found it difficult to understand the question, the item was missing for more than half of the participants in this study (51.1%). According to the internal consistency of the five-item despair scale, the Cronbach’s α scale was 0.71 (based on the non-missing answers; *n* = 68). When the item that was not understood by the participants was removed, the Cronbach’s α of the despair scale improved and was 0.75 (based on the non-missing answers; *n* = 68). Therefore, we decided to remove this item and conduct the analyses with the four-item scale.

Based on the whole sample (*n* = 139), both subscales have acceptable rates of internal consistency in this study, with Cronbach’s α = 0.75 for integrity and Cronbach’s α = 0.69 for despair.

#### 2.3.3. Depressive Symptoms

Depressive symptoms were examined with the Dutch nursing home version of the Geriatric Depression Scale [77]. The dichotomous yes/no scale with eight items assesses whether or not people feel bored, happy, helpless, hopeless, etc. The positively formulated items were reversed in order to calculate a total score of depressive symptoms, ranging from zero to eight. The higher the total score, the more depressive symptoms occur. The Cronbach’s α for this sample was 0.72, showing good internal consistency.

#### 2.3.4. Religious Background

Four single items were included in the questionnaire about patients’ religious backgrounds. These questions were asked to obtain more insight into geriatric patients’ religious background. Firstly, the importance of religion in people’s lives was assessed on a scale from not important (1) to very important (4). Secondly, participants were asked if they call themselves Catholic, Christian, Protestant, Muslim, Jewish, Humanist, none of these categories, or something else. Thirdly, two questions were asked about how often people pray and how often they are involved in religious activities, with response categories from never (1) to daily (5).

#### 2.3.5. Sociodemographic Variables

Patients’ age, gender, civil status, nationality, length of stay in the hospital, and level of education were questioned. Categories of civil status were single, married, in a relationship, widowed, and divorced. After the data were collected, categories of civil status were recoded into broader categories, namely married/in a relationship, widowed/divorced, and single. Response options regarding level of education were primary school (1), secondary education until fourteen/sixteen years old (2), secondary education until eighteen years old (3), and higher education with university or college degree (4).

### 2.4. Data Analyses

Firstly, we checked for missing data patterns and nesting of the data. Three answers were missing on the integrity scale, one answer was missing on the Geriatric Depression Scale, five answers were missing on the Brief PRCOPE scale, and ten answers were missing on the Brief NRCOPE scale. The limited missing data were missing in a random way and imputed by the use of the Expectation Maximization (EM) algorithm. The EM-method is an iterative procedure that produces maximum likelihood estimates in order to impute missing values [78]. Together with missingness patterns, it was tested whether participants were nested within hospital settings. The Intraclass Correlation Coefficient was zero for the outcome variable; thus, hospital settings were not responsible for variation in the outcome variable.

Secondly, some preliminary analyses were performed before testing the mediation model. Descriptive statistics were calculated. Correlation analysis, independent sample *t*-test, and univariate analysis of variance with post-hoc Tukey tests were used to test group differences and correlations with depressive symptoms. If background variables were of influence on depressive symptoms in this study, these were included in mediation analyses as control variables.

To test the mediation model, the PROCESS v.4 macro by Andrew F. Hayes [79,80] was used. This macro has proved to be a reliable instrument to test statistical mediation and resembles results from structural equation modeling [80]. Integrity and despair were implemented as parallel mediators (model 4 in PROCESS macro), positive and negative religious coping as separate predictors, and depressive symptoms as the outcome variable. This means that two mediation models were tested: one with positive religious coping and one with negative religious coping. Parallel mediation presupposes that two variables function as an underlying and explanatory mechanism in the relationship between two constructs [79]. The assumption of multicollinearity for the mediation variables was tested beforehand. All analyses were performed in SPSS 27.0.

## 3. Results

### 3.1. Preliminary Analysis

Table 1 displays the means, standard deviations, and correlations for each main study variable. Positive and negative religious coping were not significantly correlated with each other. Integrity and despair were negatively correlated. Both significantly correlated with depressive symptoms, but in the opposite direction. Negative religious coping negatively correlated with integrity and positively with despair and depressive symptoms. No significant correlation was found between positive religious coping and integrity, nor with despair or with depressive symptoms.

No significant differences in depressive symptoms were found between men and women. Additionally, no significant age differences, nor differences according to civil status or length of hospital stay were revealed. Further, none of the religious identifiers were significantly related to depressive symptoms. Correlation analyses showed that the level of education and depressive symptoms were negatively correlated, *r* = −0.19, *p* < 0.05. Therefore, level of education was included in the mediation model as a control variable.

### 3.2. Mediation Analysis

The first mediation model tested whether the psychosocial crisis of integrity and despair mediated the relationship between positive religious coping and depressive symptoms, controlling for educational level. All effects are presented as unstandardized effects. No significant direct or indirect pathway was found for the relationship between positive religious coping and depressive symptoms through integrity and despair (Figure 1). The bootstrap confidence interval for the total indirect effect based on 5000 bootstrap samples included zero, *B* = 0.0116, 95% CI [−0.023, 0.044]. This implies that no indirect effect was detected. Only integrity and despair were significantly related to depressive symptoms in, respectively, a negative and positive way.

The opposite is true for the second mediation model with negative religious coping as the predictor. Analyzing the indirect effect, results reveal that integrity and despair significantly mediate the relationship between negative religious coping and depressive symptoms, *B* = 0.035, 95% CI [0.005, 0.086] for integrity, and *B* = 0.061, 95% CI [0.017, 0.118] for despair. As displayed in Figure 2, negative religious coping significantly predicted integrity/despair, and the effect of integrity/despair on depressive symptoms was also significant. This was approved by the bootstrap confidence interval based on 5000 bootstrap samples, which was fully above zero and thus confirms an indirect pathway, *B* = 0.097, 95% CI [0.042, 0.174]. Additionally, a positive direct effect was found between negative religious coping and depressive symptoms when accounting for the control variables and the mediating role of integrity and despair.

## 4. Discussion

Our findings partly confirm our hypotheses. Firstly, the results show that negative religious coping statistically predicts depressive symptoms when accounting for experiences of integrity and despair, which does not apply for positive religious coping. According to the second hypothesis, analyses confirmed that how people value their past lives in terms of experiences of integrity and despair is associated with current depressive symptoms. In line with the third exploratory hypothesis, this study suggests the mediating role of the psychosocial crisis of integrity and despair in the association between negative religious coping styles and depressive symptoms. This link was not found for positive religious coping. Since this study is cross-sectional, it can be that the significant relationships found in this study function in the other direction.

The results in this paper are in line with previous research showing that religious people who struggle with their relationship with God, and with religion in general, report more depressive symptoms and vice versa [16,20,49]. Additionally, the absence of a direct link between positive religious coping styles and depressive symptoms is in line with earlier findings stating that only occasionally positive religious coping styles are related to depressive symptoms [16,26,49]. Further, with regard to the relationship between experiences of integrity and despair and depressive symptoms, our findings are confirmed by other studies. Similar to previous research, integrity is negatively related to depressive symptoms, and despair is positively related to depressive symptoms [69,71,72,76]. In other words, the general trends in previous research on the relationship between positive/negative religious coping or integrity/despair and depressive symptoms are reaffirmed in this study with geriatric inpatients.

More exploratory, this study indicates that geriatric participants maintaining negative religious coping styles suffer from more experiences of despair and have diminished experiences of integrity, which plays a role in the relationship between negative religious coping and depressive symptoms. This means that the relationship found between negative religious coping and depressive symptoms can be explained through the processes of integrity and despair. More specifically, the use of more negative religious coping seems to adversely affect how people deal with the developmental processes of integrity and despair, which is related to more depressive symptoms. Or conversely, people who report more depressive symptoms may experience more difficulties handling the developmental processes of integrity and despair and maintain more negative religious coping styles. However, that does not mean that only experiences of integrity and despair play a role in this relationship. There are also other mediators found to be significant in the relationship between negative religious coping and depressive symptoms. For example, a study in Belgium with hospitalized patients pointed out that negative religious coping styles are related to depressive symptoms through the process of hope [81]. The same was found in a study conducted in Michigan with patients undergoing major cardiac surgery [27]. Based on our results and these studies, we may assume that the relationship between negative religious coping styles and depressive symptoms is mediated by constructs such as hope, integrity, and despair. Nevertheless, further research is needed to explore other potential mediators between negative religious coping styles and depressive symptoms.

On the contrary, a significant association between positive religious coping styles and depressive symptoms through the mechanisms of integrity and despair was not found in this study. One of the reasons could be that positive religious coping is mostly associated with positively formulated constructs, not with negatively formulated constructs such as depressive symptoms. This is explained in the limitation section. Despite the lack of association between both, some other scholars do show a mediated pathway between positive religious coping and depressive symptoms. Ai and colleagues’ study [27] reported that positive religious coping styles were indirectly related to depression through hope and social support. Similarly, in Zwingmann and colleagues’ study [29], depressive coping is an underlying mechanism in the relationship between positive religious coping styles and depression.

Nevertheless, the question remains whether there are also substantive reasons for the lack of association between positive religious coping and depressive symptoms. This paper provides two possible explanations for the lack of a link between positive religious coping styles and depressive symptoms in this sample. First of all, as mentioned at the beginning of this paper, the role of religion in secularized contexts such as Belgium has to be taken into account. In Belgium, a large group of older people call themselves Christian or Catholic, but only few actively integrate religion into their daily lives. This implies that maintaining positive religious coping styles does not mean that this coping strategy is actively employed in times of crisis. As Park argues, positive religious coping styles may be less helpful for people who are not strongly religious or do not actively integrate religion in their daily life [5]. Secondly, an understanding of the difference between trait-like and state-like styles might help differentiate the difference between positive and negative religious coping styles. The trait-like style refers more to the enduring character of a person, whereas the state-like style refers more to concrete situations. It is reasonable that positive religious coping styles are dispositional and thus more trait-like compared to negative religious coping styles, which are more state-like and more related to the “crisis-specific struggle” [27] (p. 878). For example, the feeling that God punishes people through their illness (negative religious coping style) is more illness-specific and more situational than looking for a stronger connection with God in general (positive religious coping style), regardless of the specific situation. This suggests that positive religious coping styles might be not that sensitive to changing circumstances, such as illness and hospital stay, and can be interpreted as a trait aspect in people. Nevertheless, further research is needed into what may be the reasons for why positive religious coping is often not related to depressive symptoms.

Based on our preliminary analyses, two important findings can be added to our main results. Firstly, study results showed that there is no significant change in depressive symptoms according to how important religion is in people’s lives, how often they pray, or how often they are involved in religious activities. In other words, it is not the amount of religiosity or religiousness that is related to depressive symptoms in this paper, but rather how geriatric patients use religion as a coping style [27]. Secondly, our premise that positive and negative religious coping styles are separate constructs still holds true. In this study, positive and negative religious coping did not correlate with one another, and their relationship with depressive symptoms was different. This means that it remains useful to differentiate between helpful and harmful ways of religious coping [19,82].

To conclude, it is of interest to reflect on the value of the Brief RCOPE questionnaire including positive and negative religious coping styles. Pargament’s Brief RCOPE instrument [16] has often been criticized for its specific representations of God. The items of the questionnaire present God as a personal God and do not include the perception of God as nonpersonal or unknowable [83,84,85]. Especially in secularized West European countries, images of God become more nonpersonal, and the idea that God can interfere in people’s lives begins to wane [86]. For example, the item “Decided the devil made this happen” was not applicable and not recognizable for most participants in this study. Despite the fact that personal and relational images of God are known among older patients [83], it can be questioned whether older adults today still feel affiliated with these perceptions of God, or whether they feel more affiliated with nonpersonal images of God. Further research on religious coping styles needs to acknowledge the variety of personal and nonpersonal images of God among older people [17,83]. Finally, it has to be recognized that this study used religious coping styles as they are validated in the Brief RCOPE questionnaire. However, it can be interesting, in further research, to investigate how religious practices such as prayer, rituals, and participating in communities could function as forms of religious coping strategies.

## 5. Healthcare Implications

This study sheds a new light on the relationship between positive and negative religious coping styles and depressive symptoms through the psychosocial crisis in late life. This at least hints that experiences of integrity and despair in late life linked to this crisis need to be integrated in the care of older patients. This means that, as a healthcare worker, it is valuable to talk with geriatric patients about their experiences of despair, such as failures in past life that they struggle with, mistakes they made in the past that they ruminate over, and regrets about things in the past that they cannot forgive themselves for. Similarly, talking with geriatric patients about their experiences of integrity can offer more insight into how patients look back on their lives. For example, questions might include what kind of life experiences they are proud of, what kind of life events in their past lives make them feel good today, or how they try to accept missed opportunities. By giving recognition to these experiences of integrity and despair in the care of older patients, their whole life is taken into account.

In addition, a more extended and therapeutic intervention such as life review therapy can help geriatric patients in resolving the psychosocial crisis of integrity and despair. Life review is a kind of systematic reflection that evaluates the positive and negative aspects of past life [87]. During life review, people investigate what kind of feelings arise when reflecting on both good and bad memories from the past. The therapy helps people to recognize their experiences of integrity and despair, as described by Erikson [63]. Especially when geriatric patients struggle with depressive symptoms, life review is of interest [88]. The more people engage in life review, the more depressive symptoms diminish. In a study with Dutch older adults with moderate depressive symptomatology, life review therapy was effective in reducing depressive symptoms, even at three- and nine-month follow-ups [89]. In other words, the therapeutic intervention of life review succeeds at including experiences of integrity and/or despair, which often occur in late life, while having an effect on depressive symptoms.

According to the framework of person-centered care, the whole healthcare team needs to pay attention to positive and negative religious coping styles and the related depressive symptoms and experiences of integrity and despair. This specific attention can be considered a part of spiritual care. Spiritual care is one of the components of the biopsychosocial-spiritual care model, in which each caregiver needs to address the spiritual domain as a generalist [90,91]. This means that each caregiver needs to have the general skills to assess spiritual needs, be aware of spiritual themes, and be able to refer to the chaplain if needed [92]. The chaplain, who is the specialist in spiritual care and part of the healthcare team, meets all kinds of spiritual needs and themes during spiritual care interventions. The chaplain and the healthcare team work together in tandem and seek to provide the best possible spiritual care, with each having their own expertise and responsibilities. Therefore, training and education of caregivers focusing on spiritual care are required [93].

For example, when it appears that negative religious coping styles are strongly present in geriatric patients, it is appropriate to refer them to the hospital chaplain. It is the responsibility of the chaplain to ascertain the way in which negative religious coping styles are present—i.e., chronically or temporarily present—and whether or not these are related to depressive symptoms and the psychosocial crisis in late life. Additionally, positive religious coping styles need to be taken into account by the healthcare team. Positive religious coping styles are used more frequently compared to negative religious coping styles and can help religious geriatric patients deal with their situation in the hospital. Positive religious coping can be supported by staff members, especially chaplains.

## 6. Limitations

Six limitations have to be taken into account.

Firstly, since this study is cross-sectional and the mediation model is exploratory, findings should be interpreted with caution. Due to the cross-sectional design, no causal interpretations can be made. In this study, it was assumed that positive/negative religious coping styles theoretically predicted depressive symptoms, but the reverse direction can also be true. Additionally, no changes over time can be detected. For example, a study with Belgian nursing home residents demonstrated that the psychosocial crisis in late life can transform over time within people [68]. Similarly, the differences between the use of negative religious coping styles on the long and short term cannot be identified. Therefore, longitudinal research is needed to identify the causal interpretation of this relationship.

Secondly, it is reasonable that important confounding variables are missing in our mediation model. In this study, we have included educational level in our mediation model, as this was related to depressive symptoms. Unmeasured variables, such as social support and loss experiences, may play a role in the relationship as well and need to be further investigated [94,95].

Thirdly, the study can be biased by the fact that questionnaires were administered orally. It is possible that the answers given by the patients are biased by social desirability. This means that people have a tendency to respond in such a way that conforms to what is considered socially acceptable [96]. For example, it is not unlikely that people pretend to be less depressed than they really are. Although it was pointed out during the interview that people were allowed to answer honestly and that the study was anonymous, this bias cannot be completely eliminated.

Fourthly, the sample size in this study is small (*n* = 139). As discussed in our preregistration (https://osf.io/wdnq6/), we aimed for a sample size of 200 participants. Once the data collection started, we found that it was infeasible to reach the number of 200 religious geriatric patients who were eligible to answer the RCOPE questionnaire. Due to the smaller sample size and the reduced statistical power, some effects may not be detectable. Therefore, these results should be considered with caution, and further research needs to include larger samples. Moreover, most of the participants in this small sample were Christians or Catholics. This limits the generalizability of our findings to other religious groups, cultures, and/or countries. Especially regarding the use of positive and negative religious coping strategies, the religious, cultural and geographical background of the Belgian geriatric participants must be taken into account [60]. For example, minority older adults in the United States (mostly African Americans) tend to use more positive religious coping strategies (*M* = 24.0, *SD* = 4.26) compared to our sample [20]. Further research on the prevalence of religious coping strategies among minority groups would be interesting.

Not included in this study is the role of secular coping strategies as explored in Zwingmann and colleagues’ study [29]. These scholars found that the relationship between positive and negative religious coping styles, on the one hand, and anxiety and depression, on the other hand, was mediated by nonreligious scoping styles. It would be of interest to replicate these findings in a sample of older adults to gain more insight into the role of secular coping strategies in late life. Therefore, a validated measurement is needed that includes both kinds of coping strategies. Recently, Riegel and Unser added secular coping strategies to the existing RCOPE scale, which turned out to be a promising instrument and needs to be further developed [8].

The last limitation concerns the lack of positively formulated outcomes in this study. Only associations with depressive symptoms were tested. Previous studies indicated that positive religious coping styles are mostly related to positively formulated outcomes such as positive affect, well-being, and life satisfaction, and less to negatively toned outcomes such as anxiety, depressive symptoms, and negative affect [12,26,49]. Further research with geriatric patients on the associations of positive and negative religious coping styles needs to include both negative and positive health outcomes.

## 7. Conclusions

This study provides new insights into the relationship between positive and negative religious coping styles and depressive symptoms through their relationship with experiences of integrity and despair. For positive religious coping styles, no direct or indirect relationship with depressive symptoms was found. For negative religious coping styles, however, significant relationships were found. The more geriatric patients use negative religious coping styles, the more they report depressive symptoms, and vice versa, through the mechanisms of integrity and despair.

Despite the fact that the public role of Christianity in West European secularized societies is changing, it is still useful to be aware of positive and negative religious coping strategies in geriatric care, especially when they occur in combination with depressive symptoms and/or the late-life psychosocial crisis. Spiritual care for geriatric patients suffering from negative religious coping styles and/or depressive symptoms needs to ensure that experiences of integrity and despair are recognized.

Further research is needed on the role of religious coping styles in geriatric wards that takes into account the contemporary diversity of God images and religious beliefs.

## Figures and Tables

**Figure 1 ijerph-19-03835-f001:**
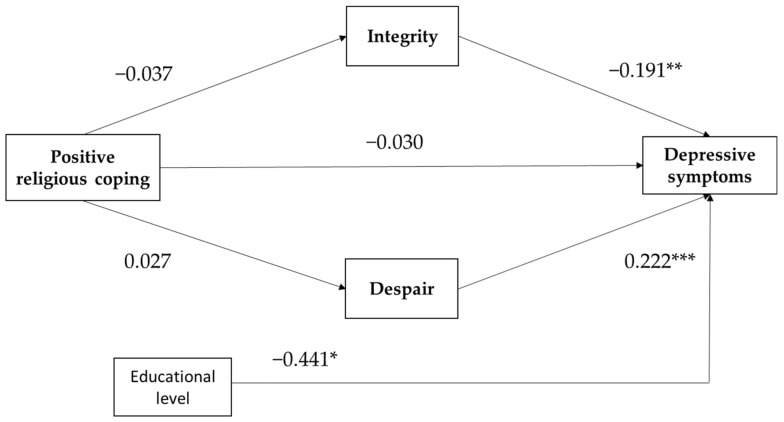
Path model of positive religious coping with depressive symptoms, and with integrity and despair as parallel mediators. Educational level is control variable. All coefficients are unstandardized. * *p* < 0.05, ** *p* < 0.01, *** *p* < 0.001.

**Figure 2 ijerph-19-03835-f002:**
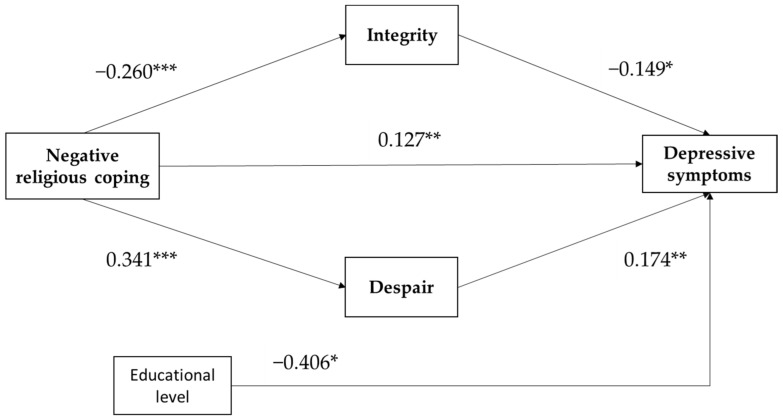
Path model of negative religious coping with depressive symptoms, and with integrity and despair as parallel mediators. Educational level is control variable. All coefficients are unstandardized. * *p* < 0.05, ** *p* < 0.01, *** *p* < 0.001.

**Table 1 ijerph-19-03835-t001:** Descriptive statistics and correlations for main study variables.

Variable	*M* (Potential Range)	*SD*	*M* Total Score (Potential Range)	*SD*	1	2	3	4	5
1. Positive RCOPE	2.54 (1–4)	0.80	17.78 (7–28)	5.60	—				
2. Negative RCOPE	1.54 (1–4)	0.50	10.77 (7–28)	3.52	−0.06	—			
3. Integrity	3.22 (1–4)	0.63	12.89 (4–16)	2.54	−0.09	−0.36 **	—		
4. Despair	1.81 (1–4)	0.73	7.24 (4–16)	2.92	0.05	0.41 **	−0.50 **	—	
5. Depressive symptoms	0.24 (0–1)	0.24	1.96 (0–8)	1.91	−0.03	0.42 **	−0.43 **	0.46 **	—

** *p* < 0.01.

## Data Availability

The data presented in this study are available on request from the corresponding author.

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
