# Peer review of "Religious Coping Styles and Depressive Symptoms in Geriatric Patients: Understanding the Relationship through Experiences of Integrity and Despair"

_ijerph, 2022, doi:10.3390/ijerph19073835_

Round 1
Reviewer 1 Report
All in all, this is a good paper, well presented with a robust methodology. As an expert, I have a couple of reservations, considering the ways the Authors refer to concepts. First, the Authors talk about "religion" at large but the issue of the diversity of religions and their impact on psychological coping is not clearly adressed. The A. are right to poin that religious coping can be positive or negative but they end up the discussion on only one of these two aspects, and it is not clear to me how religion can inflect psychological balance (by means of rituals ? beliefs ? symbols ?) and the complexity of the relation between coping styles and depression is well in place but would benefit from a broader discussion (what in the case the link is NOT that clear ?) By the way, the concept of "style" has receive little definition, even if it's only a descriptive one. same remark for the two concepts of "integrity" and "despair" taken for granted whilst they should require a longer discussion on the way they are expressed. Further, the research seems to be based on the assumption that religious belonging is mechanically linked to religious meaning / belief. Is this the A. point of view ? and finally, a comparison with SECULAR or PROFANE forms of coping would strenghten the intellectual position held by the paper (and should explain what is specifically RELIGIOUS in a religious coping).
That said, the paper is quite solid and deserves a publication, but the Authors would wisely respond to these queries and ensure robustness to their demonstration.
Reviewer 2 Report
Religious coping styles and depressive symptoms in geriatric patients. Understanding the relationship through experiences of integrity and despair
The study examines the association between religious coping styles (positive and negative) and depression for a sample of adults aged over 69 years. It specifically explores the mediating role of experiences of integrity and despair. It is a significant, well-written and well-structured study, with a careful and appropriate methodological design. The introduction provides a clear and very complete explanation of the findings from an analysis of the theoretical and empirical background. The study aims and hypotheses are clearly formulated and significant for the generation of meaningful knowledge regarding the topic in question. The data analysis is adequate and uses proven strategies and techniques. The use of the PROCESS macro for SPSS is notable. Both the discussion and the conclusions have been produced based on the findings and offer ideas about their interpretation. The highly extensive limitations section is notable; it clearly and exhaustively identifies the study limitations. In summary, this article concerns research that has been conducted well and adds significant knowledge based on a good design.
There are some elements that the authors might consider clarifying in order to improve the quality of the article:
The first element is highly important. The sample size is small (139 individuals), which could raise doubts concerning statistical power. I am concerned that the sample size is insufficient to correctly identify a genuine effect. From my perspective, the authors should address this point in the article, showing the results of the calculations of the minimum sample size based on their data analysis technique. I would like to stress the importance of this issue, since it clearly and evidently influences the quality of the findings and, by extension, the whole article.
The second element is related to the description of the Brief RCOPE. Specifically, it would be useful to know if this scale has been validated among older adults, and if it has not, to argue its suitability for that population.
The third element concerns the removal of an item of the despair scale. It is not generally appropriate to eliminate items from a scale. The authors could address the impact of this elimination on the psychometric properties of the instrument in more detail.
Fourth, the authors state (page 10, row 435) that the results did not show an association between depressive symptoms and measures of religious background. However, these variables might help to explain the failure to find a mediating role for negative religious coping with symptoms of depression. I wonder if the authors have included these variables (religious background) as control variables (in the same way as education is included) in some analyses, event if those analyses were not shown in the paper.
I am grateful for the opportunity to review this article and I congratulate the authors on their work.
Reviewer 3 Report
This study attempts to assess the relationships between religious coping and depressive symptoms, establishing a mediation model through the variables of integrity and despair. The results have interesting implications for the field of mental health, improving the understanding of the relationships between the mentioned variables.
Introduction section
The introduction is extensive and provides an adequate theoretical framework for understanding the research carried out. In addition, the bibliography used for it is very relevant and helps to give importance to the need for a study of these characteristics. It also contextualizes the study through data on the particular situation of the country where the research is carried out. In any case, the following are a few suggestions.
Perhaps to briefly introduce religious coping, it may be useful to talk about coping in general first. For this purpose, it may be appropriate to mention the work of Lazarus and Folkman (1986), the first exponents in this field. The main reason for this is so that readers unfamiliar with the subject may understand that there are different coping styles or strategies, and that religion is one of them (and perhaps one of the most important).
In this sense, to highlight the importance of the study of religious coping in the mental health field, it may be useful to mention Koenig's (2009, p. 285) reasons for the use of this type of coping. (https://doi.org/10.1177/070674370905400502)
On the other hand, it might be interesting to mention some reference to studies on religious coping and depression in different age ranges to compare between them. It may help to better understand the situation of this relationship in old age. Some studies on this topic at different ages are:
https://doi.org/10.1007/s10943-017-0359-3
https://doi.org/10.1093/jpepsy/jsu011
https://doi.org/10.1007/s10943-021-01185-x
https://doi.org/10.1111/jocn.14113
When introducing integrity and despair as mediating factors between religious coping and mental health, it may be interesting to mention the work of Koenig et al. (2012), who proposed a number of factors that may be mediating this relationship. (Koenig, H. G., King, D. E. and Carson, V. B. (2012). Handbook of religion and health (2nd ed.). Oxford University Press)
Finally, to aid the reader's understanding, it may be useful to state the hypotheses related to each objective after the objectives, rather than stating all the objectives first and then all the hypotheses.
Method and materials section
The information presented in this section is complete and allows a clear understanding of the nature of the research.
As a recommendation, the order of the subsections could be changed. The participants and the instruments should be presented first, followed by the procedure and the data analysis.
With respect to the participants, it would be appropriate to present the information in a more general manner. Specific data such as percentages should be presented in the results, in a section on the sociodemographic characteristics of the sample.
Regarding the measurement of religious background, it would be appropriate to present information on the psychometric properties of the scale used.
Results section
The results are also shown in a clear and orderly manner, both in the text and in the tables and figures.
As mentioned above, it may be appropriate to include a section to present more extensively (in text or table format) the sociodemographic characteristics of the participants.
Discussion section
This section perfectly summarizes the findings and clearly relates them to previous research. In addition, they provide a number of implications and limitations that highlight the importance of this study and the need for further research on the relationships between religion/spirituality and mental health.
Perhaps, in the sub-section on implications, it may be interesting to add some references about the need to add religion and spirituality to the education of healthcare personnel.
https://doi.org/10.2105/AJPH.2016.303501
https://doi.org/10.1037/scp0000195
https://doi.org/10.1016/j.jpainsymman.2016.05.018
Finally, I congratulate the authors on their study, as it is very comprehensive and represents an important advance in the understanding of the relationship between R/S and mental health, with important theoretical and practical implications.
